# MarvelD3 Is Upregulated in Ulcerative Colitis and Has Attenuating Effects during Colitis Indirectly Stabilizing the Intestinal Barrier

**DOI:** 10.3390/cells11091541

**Published:** 2022-05-04

**Authors:** Franziska Weiß, Carolina Czichos, Lukas Knobe, Lena Voges, Christian Bojarski, Geert Michel, Michael Fromm, Susanne M. Krug

**Affiliations:** 1Clinical Physiology/Nutritional Medicine, Charité—Universitätsmedizin Berlin, 12203 Berlin, Germany; franziska.weiss@charite.de (F.W.); carolina.czichos@gmail.com (C.C.); lukas.knobe@charite.de (L.K.); lena.voges@charite.de (L.V.); michael.fromm@charite.de (M.F.); 2Department of Gastroenterology, Rheumatology and Infectious Diseases, Charité—Universitätsmedizin Berlin, 12203 Berlin, Germany; christian.bojarski@charite.de; 3Transgenic Technologies, Charité—Universitätsmedizin Berlin, 13125 Berlin, Germany; geert.michel@charite.de

**Keywords:** MarvelD3, tight junction, inflammatory bowel disease, ulcerative colitis, IL-13, DSS-induced colitis

## Abstract

In inflammatory bowel disease (IBD), the impaired intestinal barrier is mainly characterized by changes in tight junction protein expression. The functional role of the tight junction-associated MARVEL protein MARVELD3 (MD3) in IBD is yet unknown. (i) In colon biopsies from IBD patients we analyzed MD3 expression and (ii) in human colon HT-29/B6 cells we studied the signaling pathways of different IBD-relevant cytokines. (iii) We generated a mouse model with intestinal overexpression of MD3 and investigated functional effects of MD3 upregulation. Colitis, graded by the disease activity index, was induced by dextran sodium sulfate (DSS) and the intestinal barrier was characterized electrophysiologically. MD3 was upregulated in human ulcerative colitis and MD3 expression could be increased in HT-29/B6 cells by IL-13 via the IL13Rα1/STAT pathway. In mice DSS colitis, MD3 overexpression had an ameliorating, protective effect. It was not based on direct enhancement of paracellular barrier properties, but rather on regulatory mechanisms not solved yet in detail. However, as MD3 is involved in regulatory functions such as proliferation and cell survival, we conclude that the protective effects are hardly targeting the intestinal barrier directly but are based on regulatory processes supporting stabilization of the intestinal barrier.

## 1. Introduction

Inflammatory bowel disease (IBD), comprising ulcerative colitis (UC) and Crohn’s disease (CD), is a group of chronic idiopathic relapsing and remitting gastrointestinal autoimmune disorders featuring increasing incidences over a few decades [1,2].

Although IBD can occur at almost any age, most patients obtain their diagnosis between their thirties and fourties [3]. Due to the lack of a cure, the treatment for IBD aims at gaining and maintaining remission, and preventing complications.

IBD is considered multi-factorial, as genetic factors in parallel with environmental, infectious, and immunologic factors have been identified. Dysregulation of intestinal barriers as well as enhanced mucosal immune responses are involved, either by representing a consequence or a triggering cause of impairment [4]. Of special importance for the intestinal epithelial barrier is the tight junction (TJ). The TJ is built by multiprotein complexes that are located at the most apical position between epithelial cells and forms cell–cell connections that tighten the paracellular cleft and regulate transepithelial passage and barrier properties [5]. There are two types of TJs, the bicellular TJ (bTJ) which forms a belt-like meshwork between two epithelial cells [6] and the tricellular TJ (tTJ) formed by converging bTJs at points of contact of three or sometimes four cells [7].

Transmembrane proteins of the TJ can be grouped into four families: (i) the claudin family, including 27 members in mammals [8], which is mainly responsible for general barrier formation but, brought about by channel-forming claudins, also for ion and water permeability, (ii) junctional adhesion molecules [9], (iii) the angulin family, comprising three members, angulin-1, -2, and -3, all localized predominantly at the tTJ [10], and (iv) TJ-associated MARVEL (myelin and lymphocyte protein and related proteins for vesicle trafficking and membrane linking) proteins (TAMPs), comprising occludin [11], tricellulin (also named MARVELD2) [12], and MARVELD3 [13,14].

For many TJ proteins, altered expression and functional involvement in IBD has been shown. Whereas downregulation of barrier-forming claudins such as claudin-3, -4, -5, -7 or -8 [15,16] was linked to structural impairment and loss of TJ integrity, upregulation of claudin-2, a cation and water channel-forming protein, in UC [15,17] and CD [16] led to a loss of cations and water, explaining the symptom of leak flux diarrhea.

Regarding the TAMPs, changes in IBD have been reported only for occludin and tricellulin. Whereas occludin was downregulated in both IBDs [16,17], tricellulin was downregulated in UC [18]. However, in CD tricellulin was shifted from the depth of the crypts to the surface epithelium [18], which was linked to downregulation of angulin-1 [19]. From this, the tTJ and its resulting impairment in IBD could be seen as an important factor for the enhanced paracellular luminal antigen uptake that is observed in IBD.

For the third TAMP, MARVELD3 (MD3), nothing is known about a potential change and function in IBD. It has been described as a TJ protein existing in two splice variants, MD3 v1 and MD3 v2, which has an unclear function for barrier formation and seems to be more relevant in TJ stabilization [13,14], cell behavior and survival [20], and development [21,22].

As these are also factors that might play a role in the impaired epithelium in IBD, we were interested in potential changes of MD3 in IBD and regulatory processes beyond. As we found changes in MD3 expression, we analyzed the functional impact of these on the intestinal barrier using MD3 overexpression mice and studied the regulatory pathways that could explain the observed expression changes.

## 2. Materials and Methods

### 2.1. Patients Features

The study was approved by the local ethics committee (No. EA4/015/13). Eight CD patients (inflammation was detected in the sigmoid colon), eight UC patients, and eight patients without intestinal diseases who underwent colonoscopy were included in the study (Table 1). The simple endoscopic score for CD (SES-CD) is a scoring system for staging endoscopic findings of CD patients. The score considers (i) the number and size of mucosal ulcers, the extent of the (ii) affected and (iii) ulcerated surface, and (iv) the presence of a narrowed intestinal lumen [23]. The Mayo endoscopic subscore is part of the full Mayo score to access disease presentation during colonoscopy [24] and is used for UC patients. Here, a score of 2 to 5 points is stated as mild and higher than 6 indicates moderate to severe disease activity.

Criteria for exclusion from our study were: an age less than 18 or above 80 years, pregnancy, existence of severe diseases such as neoplastic diseases, other immunological diseases and chronic inflammatory diseases, treatment with biotics, presence of fistula or perforation, in need of surgery, and, of course, lack of consent for the study.

Patients without gastrointestinal diseases (e.g., IBD, irritable bowel syndrome, diarrhea, and preceding gastrointestinal surgery) were defined as healthy controls.
cells-11-01541-t001_Table 1Table 1Characteristics of the enrolled patients.CharacteristicControl (*n* = 8)UC (*n* = 8)CD (*n* = 8)Age (median, range)48, 20–7237, 23–7937, 21–61Gender (male/female)2/65/32/6Mayo endoscopic subscore (median, range)-2, 1–3-SES-CD (median, range)--5.3, 1–12

### 2.2. Cell Lines

The human colon cell line HT-29/B6 [25] was cultured at 37 °C and 5% CO_2_ with 10% fetal bovine serum (FBS) and 100 U/mL penicillin, and 100 µg/mL streptomycin (Corning, NY, USA) in RPMI medium 1640 with GlutaMAX (Gibco, ThermoFisher, Waltham, MA, USA). Cells were seeded on PCF filters (3.0 µm pore size, 0.6 cm^2^ area, Merck Millipore, Darmstadt, Germany) and grown until confluence was reached and TER values exceeded 500 Ω cm^2^.

### 2.3. Cytokines and Inhibitors Experiments

For cytokine treatment, cells were incubated with the substances listed in Table 2 for 24 h (in the case of TNFα) or 48 h (for all other cytokines). For inhibitor experiments, a 1 h pre-treatment was performed prior to cytokine treatment at listed concentrations.

### 2.4. Protein Isolation and Western Blotting 

Cells were harvested from PCF filters (Sigma-Aldrich, Schnelldorf, Germany) using a 200 µL total lysis buffer (10 mM Tris–HCl (pH 7.5), 150 mM NaCl, 0.5% (*v*/*v*) Triton X-100, 0.1% (*w*/*v*) SDS, and protease inhibitors (Complete, Roche, Mannheim, Germany)). After an incubation time of 1 h on ice, the proteins were collected from the supernatant after 15 min of centrifugation at 15,000× *g* (at 4 °C).

For analysis of tissues, samples were homogenized in an ice-cold membrane lysis buffer (1 M Tris-HCl pH 7.4, 1 M MgCl_2_, 0.5 M EDTA, 0.5 M EGTA, and protease inhibitors) using a FastPrep-24 homogenizer (MP Biomedical). Supernatants of a short centrifugation (1000× *g*, 5 min, 4 °C) were centrifuged (42,100× *g*, 30 min, 4 °C) and the resulting pellets containing the membrane protein fraction were resolved in a total lysis buffer.

After boiling with a Laemmli buffer, protein samples were separated on a 12.5% SDS polyacrylamide gel and transferred onto a PVDF membrane (PerkinElmer, Rodgau, Germany). Proteins were detected using primary antibodies, as listed in Table 2. After washing and incubation with the respective peroxidase-conjugated secondary antibodies, proteins were detected. For this, the membranes were washed and incubated in SuperSignal West Pico PLUS (ThermoFisher, Waltham, MA, USA) and imaged using a Fusion FX7 (Vilber Lourmat). Densitometric analysis was performed using AIDA (Elysia) and all values were normalized to β-actin or α-tubulin signals of the respective blot.

### 2.5. Immunofluorescent Staining 

Paraffine sections were baked at 60 °C over night and deparaffinized by washes in xylene 3 × for 20 min; rehydration slides were incubated shortly after in 100%, 100%, 95%, 80%, and 70% ethanol and, as a final step, in deionized water. Sections were boiled for 30 min in a TEC buffer (2 mM Tris, 1.3 mM EDTA, 1 mM Tri-sodium citrate, pH 7.8). After cooling down, sections were blocked with 4% goat serum in TBST for 30 min and then for another 30 min with a Dako-blocking diluent (Dako North America Inc. S3022). The primary antibody (MD3) was incubated at 4 °C over night and the secondary antibody for 1.5 h at room temperature, which was followed by incubation with ZO-1 and DAPI for another 1.5 h at room temperature. Images were taken using a laser-scanning microscope with an extinction wavelength of 488 nm, 542 nm, and 633 nm.

### 2.6. Generation of Mice with Intestinal Overexpression of MD3

A generation of MD3 overexpressing mice was performed in accordance with the German law on animal protection (Landesamt für Gesundheit und Soziales (LAGeSo), Berlin G0200/12). The vector construct targeting this region contained the coding sequence for an eGFP-tagged version of the murine MD3 and allowed the induction of expression under the control of Cre-recombinase. Additionally, a CAG-promotor was introduced to allow different expression strength (Figure 1).

The vector construct was injected into zygotes of Villin-Cre mice [27]. Mice derived from these cells were bred with Villin-Cre recombinase-expressing mice to generate mice with intestinal epithelial overexpression of the GFP-tagged MD3. These mice were backcrossed for more than ten generations to C57Bl/6NCrl (Charles River Laboratory) and genotyping for the presence of Cre (for: 5′- GAACGCACTGATTTCGACCA-3′; rev: 5′- AACCAGCGTTTTCGTTCTGC-3′; product: 201 bp) and GFP (for: 5′- ACGAACTCCAGCAGGACCATGT; rev: 5′-TGGAGTACAACTACAACAGCCACAACGT-3′; product: 250 bp) as markers for the exogenous MD3 was used for confirmation of the introduced genes’ expression.

### 2.7. Animal Housing, Handling, and Induction of DSS Colitis

All experiments were performed in accordance with the German law on animal protection (LAGeSo), Berlin G0059/13 and T0256/16. Animals used in the experiments were male mice (12 weeks old) from heterozygous breeding and genotypical wildtype littermates were used as controls. Mice were housed under standardized conditions (12 h light/dark cycle; 22–24 °C temperature; 55% ± 15% humidity *ad libitum* access to standard diet and water). Dextran sodium sulfate (DSS, TdB, Sweden) was used for induction of colitis. The amount of 3% DSS in drinking water was given *ad libitum* over 5 days. Control mice received the same water without DSS. For determination of the disease activity index (DAI), the following parameters were observed: weight loss (0 points = no weight loss or gain, 1 points = 5–10% weight loss, 2 points = 11–15% weight loss, 3 points = 16–20% weight loss, and 4 points ≥ 21% weight loss); stool consistency (0 points = normal and well formed, 2 points = soft and unformed, and 4 points = watery stool); and bleeding stool score (0 points = normal color stool, 2 points = reddish color stool, and 4 points = bloody stool). Calculation of the DAI was based on the combined scores of weight loss, stool consistency, and bleeding ranging from 0 to 12. At day 6 the mice were sacrificed, and the colon was removed. After measurements of length and width, the colon was used for electrophysiological experiments or pieces were immediately frozen in liquid nitrogen and stored at −80 °C for subsequent analysis. For histological analysis using H&E staining, colon segments were fixed with formalin and embedded in paraffin.

### 2.8. Dilution Potentials and Impedance Spectroscopy

Voltage and transepithelial resistance (TER; R^t^) were measured in Ussing chambers with an area of 0.049 cm^2^. The contribution of the bathing solution and the filter to the measured resistance was determined prior to each experiment and subtracted. Ussing chambers and water-jacketed gas lifts were filled with 10 mL of standard Ringer’s solution (in mM: Na^+^ 140; Cl^–^ 149.8; K^+^ 5.4; Ca^2+^ 1.2; Mg^2+^ 1; HEPES 10; D(+)-glucose 10; D(+)-mannose 10.0; beta-hydroxybutyric acid 0.5; and L-glutamine 2.5 mmol/L). The pH was adjusted to 7.4 with NaOH. The solution was equilibrated with 5% CO_2_ and 95% O_2_ at 37 °C. Sodium and chloride permeabilities were determined from dilution potentials and the Goldman–Hodgkin–Katz equation, as reported before [28,29,30]. Briefly, potential changes were recorded by switching the solution of one hemichamber to a solution containing a reduced concentration of NaCl, but with all other components identical to the standard Ringer’s. Osmolality was balanced by mannitol. The resulting dilution potentials were used for further calculations.

One-path impedance spectroscopy was based on a model describing the epithelial resistance, R^epi^, as a parallel circuit consisting of the transcellular resistance, R^trans^, and the paracellular resistance, R^para^. The subepithelial resistance, R^sub^, was in series with the R^epi^. Colonic tissue of mice was mounted into Ussing chambers, to which the application of alternating currents (35 µA/cm^2^, frequency range 1.3 Hz to 65 kHz) was possible. Resulting changes in tissue voltage were detected by phase-sensitive amplifiers (402 frequency response analyzer, Beran Instruments; 1286 electrochemical interface; Solartron Schlumberger). Complex impedance values were calculated and plotted in Nyquist diagrams. R^t^ (=TER) could be obtained at minimum and R^sub^ at maximum frequency and R^epi^ was calculated using R^epi^ = R^t^–R^sub^.

### 2.9. Flux Measurements

All flux studies were performed in Ussing chambers under short-circuit conditions. Dextran fluxes were measured in 5 mL of circulating Ringer’s solution. After the addition of 100 μL of 20 mM FITC-labeled dialyzed dextran (4-, or 10-kDa FITC-dextran; TdB) to the mucosal side and unlabeled dextran of the same concentration on the serosal side, serosal samples (300 μL) were collected and replaced by fresh solution at 0, 30, 60, 90, and 120 min. For the measuring of fluorescein fluxes, Ussing chambers were filled with 10 mL of Ringer’s per side, 10 μL of fluorescein (100 mM) was added on the mucosal side, and serosal samples (300 μL) were collected and replaced 0, 10, 20, 30, and 40 min after addition. Tracer fluxes were determined from the collected samples by measuring the concentrations with a fluorometer at 520 nm (Tecan Infinite M200, Tecan, Switzerland).

For quantifying the passage of horseradish peroxidase (HRP), 100 μL of 20 μM HRP (HRP type VI, Sigma-Aldrich) was added on the mucosal side and serosal samples of 300 μL were taken and replaced by fresh solution 0, 30, 60, 90, and 120 min after addition. Samples and standards were analyzed using a fluorogenic peroxidase substrate (QuantaBlu, Pierce, Bonn, Germany) for reaction and measured in a fluorometer (Tecan Infinite M200, Tecan, Switzerland) using wavelengths of 325 nm (excitation) and 420 nm (emission).

### 2.10. Statistical Analysis

Data were expressed as mean values ± standard error of the mean (SEM) if not stated otherwise. Statistical analyses were performed using Student’s t-test for comparison between two groups or one-way ANOVA for comparing more than two groups (multiple testing), where *p* < 0.05 was considered significant.

## 3. Results

### 3.1. Expression of MD3 in Intestinal Epithelium of IBD Patients

Analysis of membrane protein fractions from IBD patients and healthy controls revealed that MD3 v1 was upregulated in UC (control: 100.0 ± 10.4%, UC: 259.9 ± 51.3%), but only showed a tendency for increase in CD (CD: 120.6 ± 1.7%, *p* = 0.069), whereas MD3 v2 was not affected (control: 100.0 ± 1.1%, CD: 97.2 ± 11.1%, UC: 84.3 ± 8.0%; Figure 2A). Moreover, the localization of MD3 was not altered in either CD or UC patients compared with controls, as indicated by immunofluorescence staining (Figure 2B).

### 3.2. Signaling Pathways Involved in MD3 Upregulation

To analyze the responsible factors for the MD3 increase in UC, several cytokines of relevance in IBD, especially UC, were tested for their effects on MD3 expression. For this, the human colon epithelial cell line HT-29/B6 was used. Cells were treated with the cytokines listed in Table 2 for 48 h (or 24 h with TNFα). Initial tests suggested that IL-4 and IL-13 were the most promising candidates to be involved in the upregulation of MD3 in UC (Figure 3A).

Regarding IL-4, changes in UC are controversially discussed because there are reports of downregulation [31]; thus, focus was laid on IL-13.

The involvement of known signaling pathways for IL-13 in this regulation were studied using different inhibitors (Figure 3B,C). SAHA, an inhibitor of STAT6, and the STAT3 inhibitors, Inhibitor VI and AG490, were able to inhibit the effect of IL-13 on MD3 expression, indicating involvement of these two STATs. Tanshinone IIa partly reduced the effect of IL-13 suggesting that transcription AP-1 might be involved in MD3 expression, but not as a main player. Inhibition of MAPK/Erk pathways (U0126), JNK (JNKV and SP600125), and JAK3 still led to an increase in MD3 expression. These results point towards a STAT3/6-regulated signaling, which has been described to be initiated via the binding of IL-13 to the IL-13Rα1.
Figure 3**Analysis of signaling pathways regulating MD3 expression.** (**A**) Western blot images of screening for cytokines having an upregulating effect on MD3 in HT-29/B6 cells. (**B**) Densitometric analysis of MD3 expression in HT-29/B6 under the influence of IL-13 and different inhibitors. Among the tested inhibitors, STAT3 and STAT6 inhibition led to loss of the effect of IL-13 on MD3 expression (control = condition without any inhibitor pretreatment, * *p* < 0.05; ** *p* < 0.01; *** *p* < 0.0001 tested against the respective IL-13-free condition; # *p* < 0.05; ### *p* < 0.001 tested against IL-13 treatment; *n* = 3–5). (**C**) Representative Western blot images.
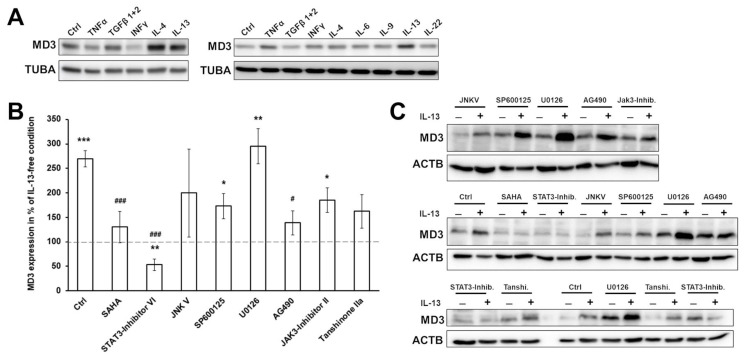


### 3.3. Effects of MD3 in DSS-Induced Colitis

To analyze the effect of MD3 in colitis, mice with intestinal overexpression of MD3 (MD3-OE), under control of the Villin-Cre promoter, were generated. The mice showed no obvious phenotypic difference to the wt mice and developed normally.

Colitis was induced by DSS application through the drinking water in male, 12-week-old mice. Male mice were used as it was shown that there are differences in the manifestation of colitis in male and female mice [32,33], and that also, intestine length and animal weight were not comparable to each other in the same age groups. In addition, due to the female hormone cycle, changes in TJ proteins in the gut have been reported [34].

DSS MD3-OE mice developed less severe colitis (lower DAI) than DSS wildtype (wt) mice (Figure 4A), which was based on a lower occurrence of bloody and watery stools, but not in body weight loss. Colon length (Figure 4B) was reduced in both wt DSS and MD3-OE DSS compared to the respective controls, whereas there was no significant difference between colon length or width of wt and MD3-OE mice. In addition, the histological appearance of the colon was not notably different in any condition (Figure 4C). In wt as well as in MD3-OE, DSS treatment led to distorted crypt architecture, loss of the superficial epithelial lining, and immune cell infiltrations. However, in MD3-OE animals the extent of these changes appeared to be less eminent, which was in line with the appearance of their stool (Figure 4C, right MD3-OE DSS image).

MD3 expression in these MD3-OE mice was approximately doubled in comparison to wt mice (Figure 5A). Expression of other TJ proteins within the intestine was unchanged. Only for occludin was an increased expression observed, which was not significant due to huge variation (Figure 5B,C).

Some TJ protein expression levels were affected by DSS in wt as well as in MD3-OE (Figure 5B,C): Regarding claudin-1, the measured increases varied massively compared to the low expression under control conditions, making these changes not significant. Claudin-4 tended to increase as well (also ns), but was less obvious than claudin-1. Under the influence of DSS, claudin-15 and tricellulin were clearly downregulated (claudin-15: wt Ctrl: 100.0 ± 8.7% *n* = 6; wt DSS: 36.3 ± 12.8, *n* = 5, * *p* = 0.049; MD3-OE Ctrl: 104.8 ± 20.4, *n* = 10; and MD3-OE DSS: 43.5 ± 9.9, *n* = 9, * *p* = 0.036; tricellulin: wt Ctrl: 100.0 ± 10.9% *n* = 5; wt DSS: 42.4 ± 15.6, *n* = 5, * *p* = 0.025; MD3-OE Ctrl: 98.4 ± 22.0, *n =* 10; and MD3-OE DSS: 50.1 ± 8.7, *n* = 11, ** *p* = 0.0099).

MD3 was upregulated in wt mice under DSS to levels comparable to the ones found in MD3-OE mice under control conditions (Figure 5B,C). In these mice, no further increase of MD3 occurred under DSS.
Figure 5**Expression of MD3 and other TJ proteins.** (**A**). Representative Western blots for MD3 and β-actin of wt mice and MD3-OE intestinal tissue. In addition to the MD3 signal in MD3-OE, some GFP-tagged MD3 was detectable. However, as the MD3 contained its start codon in addition to the N-terminal tag, amounts of untagged MD3 occurred in MD3-OE. (**B**). Representative Western blots for MD3 and other TJ proteins of wt and MD3-OE under control conditions and under the influence of DSS with corresponding β-actin signals. (**C**). Densitometric analysis of TJ expression profiles in wt and MD3-OE mice under control conditions (Ctrl) and DSS-induced colitis (*n* = 3–13; * *p* < 0.05; ** *p* < 0.01 compared to wt Ctrl; # *p* < 0.05 compared to MD3-OE Ctrl).
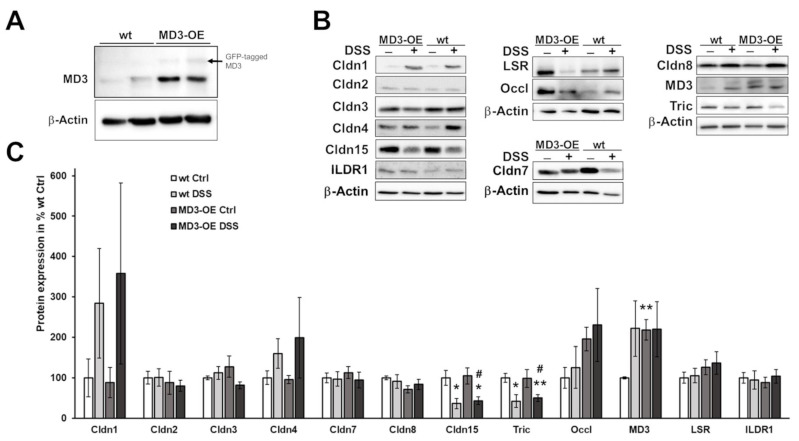


An electrophysiological barrier examination of the colonic tissue employing one-path impedance spectroscopy (Figure 6A) showed that wt and MD3-OE under steady-state conditions did not differ in their barrier properties. As expected, the transepithelial resistance (TER) decreased in wt mice with DSS-induced colitis, and this was caused by a drop in the epithelial resistance (R^epi^). Interestingly, in MD3-OE mice the DSS-treatment only showed a tendency for decrease in both TER and R^epi^. In comparison to the wt DSS, in MD3-OE DSS R^epi^ was higher, suggesting smaller barrier impairment. To characterize ion permeabilities in more detail, dilution potentials were performed. Charge selectivity P_Na_/P_Cl_ was not different in wt and MD3-OE mice under control as well as under DSS-induced conditions (wt Ctrl: 0.87 ± 0.05, *n* = 7; wt DSS: 0.89 ± 0.02, *n* = 5; MD3-OE Ctrl: 0.86 ± 0.04, *n* = 13; and MD3-OE DSS: 0.91 ± 0.03, *n* = 9).

Absolute permeabilities for Na^+^ as well as for Cl^-^ were determined to be increased under DSS conditions in wt mice as well as in MD3-OE mice, suggesting that there were no differences caused by the MD3-OE in these permeabilities.

Potential changes in transcellular transport, e.g., by endocytic activity, were analyzed measuring fluxes for HRP. In wt and MD3-OE mice under control as well as under DSS-induced colitis, permeability for HRP was unaffected (Figure 6C).

Furthermore, paracellular flux markers of different sizes were analyzed. Permeability for fluorescein, which is a small flux paracellular marker of 332 Da, only tended to be increased in wt mice during DSS-induced colitis, whereas such tendency was not visible in the MD3-OE mice. However, there were no obvious differences (Figure 6D). For the macromolecule 4 kDa FITC-dextran (FD4 Figure 6E), paracellular permeability was increased during DSS-induced colitis, but wt mice and MD3-OE mice showed no differences to each other. Similar increases were observed for 10 kDa FITC-dextran (FD10 Figure 6F). Interestingly, here the untreated MD3-OE mice already showed a distinct higher permeability for FD10 than the wt mice. However, this was still in a low range when compared to the increase occurring in colitis mice.
Figure 6**Characterization of functional intestinal barrier properties in wt and MD3-OE mice during DSS-induced colitis.** (**A**) Impedance spectroscopic differentiation of the TER in R^epi^ and R^sub^ (*n* = 11–13). (**B**) Permeabilities for Na^+^ and Cl- as determined by dilution potentials (*n* = 5–13). (**C**) Permeability for the transcellular flux marker HRP (*n* = 5–8). (**D**) Permeability for the small paracellular flux marker fluorescein (*n* = 8–11). (**E**) Permeability for the macromolecular paracellular flux marker 4 kDa FITC-dextran FD4 (*n* = 5–10). (**F**) Permeability for the macromolecular paracellular flux marker 10 kDa FITC-dextran FD10 (*n* = 7–8). (* *p* < 0.05, ** *p* < 0.01, *** *p* < 0.001 compared to wt under Ctrl conditions; # *p* < 0.05 compared to wt DSS; + *p* < 0.05, ++ *p* < 0.01 compared to MD3-OE Ctrl).
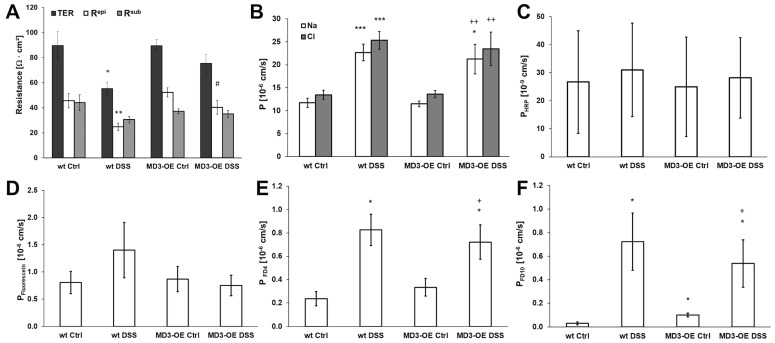


## 4. Discussion

MD3 is a member of the TAMP family, in addition to occludin and tricellulin.

Comparison of colon tissue protein samples of healthy donors with CD and UC patients revealed an upregulation of MD3 v1 in UC. In contrast, occludin [16,17] and tricellulin [18] were downregulated in CD and/or UC, respectively. A tendency for an increased expression in CD was observed, but below significance. If one assumes the regulatory functions described for MD3 in cell behavior and survival [20] and development [21,22], it might be no surprise that in both IBDs changes in MD3 could occur, even though they were more eminent in UC.

To identify the mechanism by which MD3 is upregulated, the human colon epithelial cell line HT-29/B6 was stimulated with cytokines that are typically involved in UC. Among these, IL-4 and IL-13 were able to increase MD3 expression. IL-13 is typically upregulated in UC but not in CD [17,35], which would explain the elevated expression levels of MD3 in UC but not in CD patients. On the other hand, IL-4 was found to remain unchanged or even downregulated [31].

IL-4 and IL-13 are canonical type 2 cytokines and both are able to bind to the receptor complex formed by IL13Rα1 and IL4Rα subunits. This complex is expressed in different cell types including epithelial cells [36]. Despite IL4Rα/IL13Rα1 having a higher affinity for IL-4 [37], IL-13 forms more stable complexes [38]. Although IL-4 increased MD3 expression, we did not further investigate this pathway as IL-4 is not altered in UC patients, as mentioned above. Nevertheless, as it is known that the IL-13/IL-13Rα1 complex binds with high affinity to IL-4Rα-activating STAT6 [39], a joint involvement of both cytokines may be assumed.

Our previous results showed that IL-13 induced upregulation of claudin-2 and the downregulation of tricellulin in HT-29/B6 cells [18], which could explain the leak flux diarrhea typically observed in UC and an increase in antigen uptake. Whereas claudin-2 expression is activated by the binding of IL-13 to the IL4R/IL13Rα1 receptor complex, tricellulin downregulation is induced via the IL13Rα2 receptor. Typical signaling cascades of the two receptors were analyzed using the respective inhibitors. Only the STAT3 inhibitors IV and AG490, as well as the STAT6 inhibitor, SAHA, prevented the IL-13 induced upregulation of MD3, indicating that the increased MD3 expression was regulated by the signaling pathways described for the IL13Rα1 receptor. Interestingly, the upregulation of claudin-2 can also occur via this pathway, suggesting a parallel regulation and correlation of both TJ proteins.

To investigate if MD3 upregulation has protective, or rather, inflammation enhancing effects, mice with intestinal epithelial overexpression of MD3 were generated and colitis was induced using DSS. This colitis model primarily targets the epithelium and should, therefore, help to make protective or destabilizing effects of MD3 directly visible. Furthermore, it is known that in DSS colitis colonic IL-13 expression is increased [40]. In our study, MD3 expression was enhanced in wt mice that received DSS treatment to a level comparable to that of MD3-OE mice, further supporting the link between IL-13 and MD3 upregulation. In comparison to wt mice, MD3-OE mice developed less severe colitis indicated by lower DAI values, which also could be seen in H&E staining. However, when analyzing the effects on the epithelial barrier function, MD3-OE ameliorated the DSS-induced decrease in TER and R^epi^ compared to wt DSS mice but had no influence on ion permeability or permeability for small solutes. Additionally, macromolecule permeability MD3-OE had no protective effect on the DSS-induced increase. Furthermore, as MD3 v1 was found to be especially stable at the tricellular tight junction [14], one could assume a role for MD3 here; however, in DSS-induced colitis the additional MD3 also seemed not to stabilize the tricellular TJ in the context of the macromolecule passage.

In general, it was suggested that TAMPs might partially compensate for each other [14], and interestingly, the third TAMP, occludin, tended to be increased in its expression in the MD3-OE. As occludin knockout had no obvious effects on the intestinal barrier [41], it is assumed that it has more regulatory functions and barrier-stabilizing effects than direct barrier properties. The upregulation of occludin in the MD3-OE could therefore indicate that the two TAMPs might influence each other and their regulative functions. In addition, it has been observed that localization shifts of TAMPs can occur when, for example, occludin [42] is downregulated.

Our results show the upregulation of MD3 via the IL13Rα1/STAT3 and six pathways and a general protective function of MD3-OE in mice, further supporting the previous suggestion of an anti-IL13Rα2 therapy instead of targeting the IL13Rα1 [18]. Recent findings showed that UC patients could be divided into two subsets by IL-13 mRNA tissue values and that IL-13-high patients tended to be diagnosed at a younger age and have a tendency for more severe UC [43]. However, IL-13-targeting approaches thus far did not lead to sufficient effects [44,45]. This might be explained by the chosen targeting strategies, as either the attachment between IL-13 and IL4Rα was blocked (Anrukinzumab) [44] or IL-13 itself was neutralized (Tralokinumab) [45]. Both strategies aim towards the inhibition of IL-13 effects via the IL4Rα/IL13Rα1 complex or both IL-13 receptors (IL4Rα/IL13Rα1 and IL13Rα2). Based on the results of these and previous studies of our group, this inhibition would still allow for the downregulation of tricellulin, leading to increased macromolecular permeability for the former mentioned, whereas in both treatments the claudin-2 and MD3 upregulation would be prevented, which would promote inflammatory processes instead, as a claudin-2 increase leads to leak flux diarrhea that also has epithelium rinsing and protective effects, known as enteric tears [46].

Our study suggests, furthermore, a protective function of MD3 upregulation. It is known that IL-13-induced type 2 immunity through IL4Rα/IL13Rα1 [47], which suppresses type 1 and type 17 inflammatory processes [48] and promotes wound repair [49], is a function which might be supported by MD3 as its functions in proliferation and cell survival are known [20].

However, the affinity of IL-13 to IL13Rα2 is much higher than to IL4Rα/IL13Rα1 [50]. IL13Rα2 is often referred to as being a decoy receptor for IL-13, with the function to limit IL-13 availability and thereby controlling IL-13-induced signaling. Nevertheless, based on our results, IL13Rα2 blocking would lead to a more stable barrier due to stable tricellulin levels and the upregulation of claudin-2 and MD3 via IL13Rα1. Indeed, the first in vivo results were published showing that the usage of anti-IL13Rα2 in mice during DSS-induced colitis resulted in less severe disease and shorter recovery durations [51].

It was shown that mice overexpressing claudin-2 developed less severe colitis upon DSS treatment than wt mice [52], supporting the hypothesis of the “enteric tears” as a counter-regulatory process during development of the inflammation. Comparable to this, our results with MD3-overexpressing mice indicate that upregulation of MD3 in UC patients might also not promote, but rather moderate and attenuate, inflammatory processes.

While our previous results explain the importance of stable tricellulin expression in IBD [18], the data of this study could not show that MD3-OE compensates for tricellulin reduction. Only in some mice did we observe a direct protective function of MD3 for the epithelial barrier in the DSS model. Since there was only a tendency for barrier-enhancing effects, it is more likely that MD3 has other additional effects resulting in the ameliorated disease phenotype in MD3-OE mice. As already mentioned, one of these effects could be more enhancement in proliferation or cell survival.

One possibility based on this could be that MD3 might affect neighboring cells, including not only epithelial cells but also the subjacent immune cells of the lamina propria, thereby developing its protective effects in a more indirect way. Future studies analyzing such effects in more detail will be necessary to elucidate the actual role of MD3 in the amelioration of colitis.

## 5. Conclusions

This study revealed the upregulation of MD3 v1 in UC patients and identified IL-13 as a regulating cytokine for this effect. The regulation occurred via activation of IL-13Rα1-dependent pathways involving STAT3 and 6. Overexpression of MD3 in mice had protective effects in DSS-induced colitis; however, the mechanisms behind these results need further investigation. Nevertheless, these findings support our previous suggestion of targeting the IL13Rα2 receptor in the therapy of UC, as this would inhibit IL-13-induced downregulation of tricellulin while allowing upregulation of claudin-2 and MD3, thereby stabilizing the epithelial barrier.

## Figures and Tables

**Figure 1 cells-11-01541-f001:**
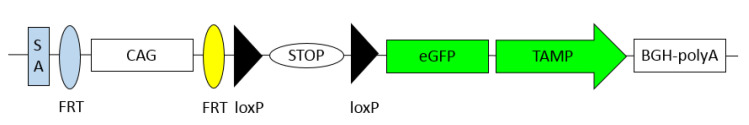
**Schematic overview of the introduced sequences coded on a vector pROSA/MD3**. SA: splice acceptor site; FRT: FRT-site for recognition by Flippase (Flp); CAG: CAG-promotor [26]; loxP: target sequence for Cre-recombinase; STOP: Stop-codon sequence; eGFP: enhanced green fluorescent protein; TAMP: cDNA sequence of murine MARVELD3 variant 2 (corresponding to the human variant 1); BGH-polyA: polyadenylation and, thereby, transcription termination site.

**Figure 2 cells-11-01541-f002:**
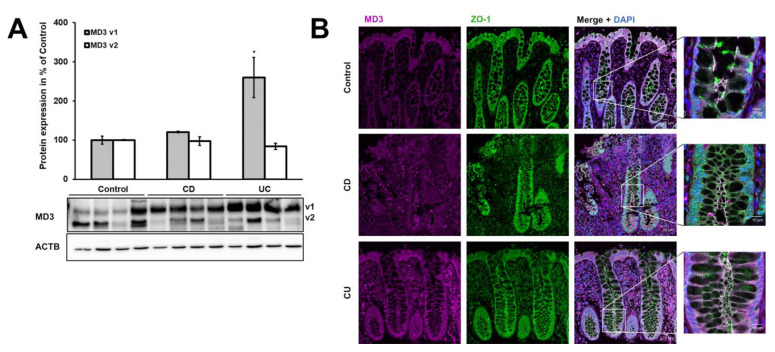
**Expression and localization of MD3.** (**A**) Representative Western blots and densitometric analysis of MD3 variants in biopsies from the sigmoid colon of control, CD, and UC patients. MD3 variant 1 (v1, upper band, ~46 kDa) was upregulated in UC, whereas variant 2 (v2, lower band, ~45 kDa) did not change (* *p* < 0.05; *n* = 8 for each group, normalized for β-actin). (**B**) Representative immunofluorescence images from colon tissue of control, CD, and UC patients to determine MD3 localization. Signal intensity was increased for better visibility and therefore could not be used for quantitative estimations. No changes in MD3 localization (magenta) were detected between controls, CD, and UC patients (counterstain ZO-1, green). Bars = 20 µm. Right images are insets of shown merge images, with higher magnification. Bar = 10 µm.

**Figure 4 cells-11-01541-f004:**
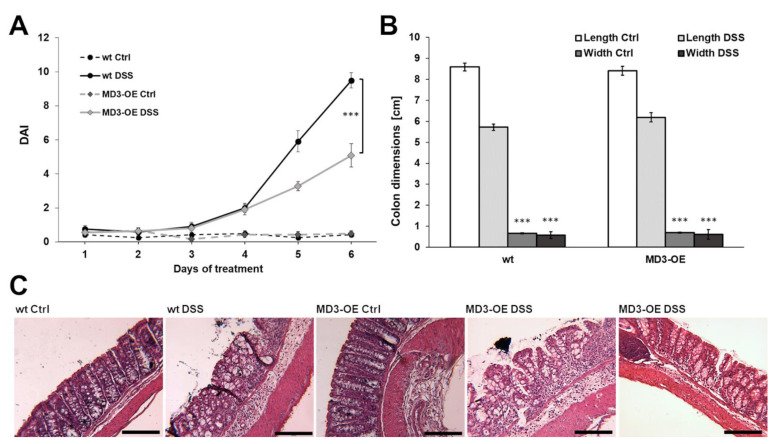
**Disease activity index and morphological changes in DSS-induced colitis.** (**A**) Disease activity index (DAI) of wildtype (wt) and intestinal MD3-overexpressing mice (MD3-OE) during DSS-induced colitis (*** *p* < 0.001; *n* = 11–12). (**B**) Length and width changes in colons of wt and MD3-OE occurred in the same manner during DSS-induced colitis (*** *p* < 0.001; *n* = 11–12). (**C**) Representative H&E stainings of colons from wt and MD3-OE mice. Notable was that in about one third of the MD3-OE mice the colon appeared to be impaired only minimally (representative image on the right; bar = 200 µm).

**Table 2 cells-11-01541-t002:** Cytokines, inhibitors, and antibodies used in this study.

	Concentration		Source
**Antibodies**			
mouse anti-ACTB	WB	1:10,000	Sigma-Aldrich, Schnelldorf, Germany, CAT. A5441
rabbit anti-CLDN1	WB	1:1000	Invitrogen, ThermoFisher, Waltham, MA, USA Cat. 519000
rabbit anti-CLDN2 (used for mouse tissue analysis)	WB	1:1000	Invitrogen, ThermoFisher, Waltham, MA, USA Cat. 516100
mouse anti-CLDN2	WB	1:1000	Invitrogen, ThermoFisher, Waltham, MA, USA, Cat. 325600
rabbit anti-CLDN3	WB	1:1000	Invitrogen, ThermoFisher, Waltham, MA, USA, Cat. 341700
mouse anti-CLDN4	WB	1:1000	Invitrogen, ThermoFisher, Waltham, MA, USA, Cat. 329400
mouse anti-CLDN7	WB	1:1000	Invitrogen, ThermoFisher, Waltham, MA, USA, Cat. 349100
rabbit anti-CLDN8	WB	1:1000	Invitrogen, ThermoFisher, Waltham, MA, USA Cat. 400700Z
rabbit anti-CLDN15	WB	1:1000	Invitrogen, ThermoFisher, Waltham, MA, USA, Cat. 364200
rabbit anti-ILDR1	WB	1:1000	Bioss, ThermoFisher, Waltham, MA, USA, Cat: bs-11013R
rabbit anti-LSR	WB	1:1000	Atlas antibodies, Cat. HPA007270
rabbit anti-MD3	IF WB	1:200 1:1000	Proteintech, Rosemont, IL, USA, Cat. 25567-1-AP
rabbit anti-MD3 (used for patient tissue and some mouse tissue analysis)	WB	1:1000	Kind gift of Jerrold Turner, Laboratory of Mucosal Barrier Pathobiology, Brigham and Women’s Hospital, Harvard Medical School, Boston, MA, USA, [13,14]
rabbit anti-OCCL	WB	1:1000	Invitrogen, ThermoFisher, Waltham, MA, USA, Cat. 711500
rabbit anti-TRIC	WB	1:1000	Invitrogen, ThermoFisher, Waltham, MA, USA, Cat. 700191
mouse anti-TUBA	WB	1:4000	Sigma-Aldrich, Schnellendorf, Germany, CAT. T9026
mouse anti-ZO-1 647	IF	1:500	Invitrogen, ThermoFisher, Waltham, MA, USA, Cat. MA3-39100-A647
DAPI	IF	1:1000	Roche Diagnositics, Mannheim, Germany, Cat. 10 236 276 001
goat anti-rabbit IgG (H+L) secondary antibody Alexa Flour 488	IF	1:500	Invitrogen, ThermoFisher, Waltham, MA, USA, Cat. A11034
goat anti-mouse IgG (H+L) antibody Alexa Flour 594	IF	1:500	Invitrogen, ThermoFisher, Waltham, MA, USA, Cat. A11032
secondary peroxidase-conjugated antibodies anti-rabbit	WB	1:10000	Jackson ImmunoResearch, Cambridge House, UK, Cat. 111-036-003
secondary peroxidase-conjugated antibodies anti-mouse	WB	1:10000	Jackson ImmunoResearch, Cambridge House, UK, Cat. 115-036-003
**Cytokines**			
TNFα	1000	U/mL	PeproTech, Hamburg, Germany
IFNγ	50	U/mL	PeproTech, Hamburg, Germany
IL-1α	10	ng/mL	PeproTech, Hamburg, Germany
TGFβ 1	10	ng/mL	Miltenyi Biotec, Bergisch Gladbach, Germany
TGFβ 2	10	ng/mL	Miltenyi Biotec, Bergisch Gladbach, Germany
IL-4	100	ng/mL	PeproTech, Hamburg, Germany
IL-5	50	ng/mL	Miltenyi Biotec, Bergisch Gladbach, Germany
IL-6	50	ng/mL	Miltenyi Biotec, Bergisch Gladbach, Germany
IL-9	100	ng/mL	Miltenyi Biotec, Bergisch Gladbach, Germany
IL-13	100	ng/mL	PeproTech, Hamburg, Germany
IL-22	100	ng/mL	Miltenyi Biotec, Bergisch Gladbach, Germany
**Inhibitors**			
U0126	10	µM	Cell signaling Technology, Frankfurt am Main, Germany
SAHA	5	µM	Sigma-Aldrich, Schnelldorf, Germany
AG490	50	µM	Sigma-Aldrich, Schnelldorf, Germany
Tanshinone IIa	10	µM	Sigma-Aldrich, Schnelldorf, Germany
STAT3-Inhibitor VI	10	µM	Calbiochem, Darmstadt, Germany
JAK3-Inhibitor II	50	µM	Calbiochem, Darmstadt, Germany
JNK V	10	µM	Calbiochem, Darmstadt, Germany
SP600125	10	µM	Cell Signaling Technology, Frankfurt am Main, Germany

## Data Availability

The data are presented in this study and are also available on request from the corresponding author.

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
