# Peer review of "MarvelD3 Is Upregulated in Ulcerative Colitis and Has Attenuating Effects during Colitis Indirectly Stabilizing the Intestinal Barrier"

_cells, 2022, doi:10.3390/cells11091541_

Round 1
Reviewer 1 Report
The manuscript entitled “MarvelD3 in inflammatory bowel diseases has attenuating effects indirectly stabilizing the intestinal barrier” by Weiß et al focuses on the role of MarvelD3, a tight junction-associated MARVEL protein, in the integrity of the intestinal epithelial barrier.
- Grammar corrections should be made in the text. Some examples are:
Line 34-35: “… the treatment for IBD aims at controlling to gain and remain remission and to prevent complications.” Should be changed for instance to “…the treatment for IBD aims at gaining and maintaining remission, and preventing complications”
Line 36-37: “IBD is considered being multi-factorial as genetic factors in parallel with environmental, infectious, and immunologic factors have been identified.” Should be changed for instance to “IBD is considered multi-factorial, as genetic factors in parallel with environmental, infectious, and immunologic factors have been identified.”
Lines 37-39: “Dysregulation of intestinal barriers as well as mucosal immune responses are involved either by representing a consequence or a triggering cause of impairment” I believe this sentence needs some editing.
Results:
“Expression of MD3 in intestinal epithelium of IBD patients”
- Figure 2: The authors should mention that the n=8 for each studied group (Control, CD and UC).
- Did the authors use an isotype control antibody for each studied factor? If not, the authors should check the specificity of their antibodies by using an isotype control antibody. If yes, they should mention it in the figure legend.
“Signaling pathways involved in MD3 upregulation”
- The authors should mention at the beginning of the first paragraph that these tests were done in HT-29/B6 cells and under what conditions (incubation hours, concentration of cytokines). Also, what is the difference of the two blots in Figure 3A? If there is no difference, authors should keep only one representative image.
- Line 252: “Tanshinone IIa partly was reducing the effect of IL-13…” should be changed to “Tanshinone IIa partly reduced the effect of IL-13…”
- Line 254-255: “Inhibition of MAPK/Erk pathways (U0126) and JNK (JNKV and SP600125) and JAK3 were still leading to an increase of MD3 expression” should be changed to “Inhibition of MAPK/Erk pathways (U0126) and JNK (JNKV and SP600125) and JAK3 still led to an increase of MD3 expression”
- Figure 3C: I believe that the authors should provide a single representative western blot image showing the control and every other inhibitor. The current Figure 3C may confuse the reader and the clear message that the STAT3/6 inhibitors are able to reduce the MD3 expression is lost.
“Effects of MD3 in DSS-induced colitis”
- I believe that the paragraph of the lines 282-291 that describes the DSS model should be moved up, at the beginning of the results, and thus Figure 5 should be change to Figure 4. As it is currently written, it gets confusing reading about the results of Figure 4, then Figure 5 and then again of Figure 4.
Author Response
Reviewer 1
The manuscript entitled “MarvelD3 in inflammatory bowel diseases has attenuating effects indirectly stabilizing the intestinal barrier” by Weiß et al focuses on the role of MarvelD3, a tight junction-associated MARVEL protein, in the integrity of the intestinal epithelial barrier.
- Grammar corrections should be made in the text. Some examples are:
Line 34-35: “… the treatment for IBD aims at controlling to gain and remain remission and to prevent complications.” Should be changed for instance to “…the treatment for IBD aims at gaining and maintaining remission, and preventing complications”
Line 36-37: “IBD is considered being multi-factorial as genetic factors in parallel with environmental, infectious, and immunologic factors have been identified.” Should be changed for instance to “IBD is considered multi-factorial, as genetic factors in parallel with environmental, infectious, and immunologic factors have been identified.”
Lines 37-39: “Dysregulation of intestinal barriers as well as mucosal immune responses are involved either by representing a consequence or a triggering cause of impairment” I believe this sentence needs some editing.
We thank the reviewer for the grammar corrections. We changed the suggested sentences and checked the remaining manuscript carefully.
Results:
“Expression of MD3 in intestinal epithelium of IBD patients”
- Figure 2: The authors should mention that the n=8 for each studied group (Control, CD and UC).
We added that n of 8 was for each group.
- Did the authors use an isotype control antibody for each studied factor? If not, the authors should check the specificity of their antibodies by using an isotype control antibody. If yes, they should mention it in the figure legend.
There were no isotope control antibodies used in this study. The antibodies used were well established already over the years in our lab and others using positive or negative controls or in some cases blocking peptides. This was done within different projects and can be found in various recent publications. As being the most important for the study, for the MD3 antibody that we received from the Turner group specificity was shown in citation [14], Raleigh et al., 2010.
“Signaling pathways involved in MD3 upregulation”
- The authors should mention at the beginning of the first paragraph that these tests were done in HT-29/B6 cells and under what conditions (incubation hours, concentration of cytokines). Also, what is the difference of the two blots in Figure 3A? If there is no difference, authors should keep only one representative image.
We agree that mentioning this at the beginning of the paragraph will be useful and added the requested information. The two blots in Fig. 3A are different preparations and we decided to show both blots, as for example in the right one would assume an effect of TNFα, which was clearly not seen in the left blot. On the other hand, the left blot would suggest that IL-4 enhances MD3 expression, while the right does not. Only IL-13 was upregulated in both. Thus, we decided to show both sample blots of the screening.
- Line 252: “Tanshinone IIa partly was reducing the effect of IL-13…” should be changed to “Tanshinone IIa partly reduced the effect of IL-13…”
We changed the sentence accordingly.
- Line 254-255: “Inhibition of MAPK/Erk pathways (U0126) and JNK (JNKV and SP600125) and JAK3 were still leading to an increase of MD3 expression” should be changed to “Inhibition of MAPK/Erk pathways (U0126) and JNK (JNKV and SP600125) and JAK3 still led to an increase of MD3 expression”
We changed the sentence accordingly.
- Figure 3C: I believe that the authors should provide a single representative western blot image showing the control and every other inhibitor. The current Figure 3C may confuse the reader and the clear message that the STAT3/6 inhibitors are able to reduce the MD3 expression is lost.
Unfortunately, providing a single representative western blot image showing the control and every other inhibitor is not possible because not all samples would fit on one blot. To keep at least some comparability between different blots, we had some cytokine treatments and control conditions loaded on other shown blots loaded as well.
We think that the message that STAT3/6 inhibitors were able to inhibit the effect of IL-13 on MD3 expression is clearly seen in the diagram that shows all data quantifications. The blots only serve as additional underlining support.
“Effects of MD3 in DSS-induced colitis”
- I believe that the paragraph of the lines 282-291 that describes the DSS model should be moved up, at the beginning of the results, and thus Figure 5 should be change to Figure 4. As it is currently written, it gets confusing reading about the results of Figure 4, then Figure 5 and then again of Figure 4.
We moved the figures as suggested and also adjusted the respective text to keep the flow.
Reviewer 2 Report
Great use of human tissue, culture and mouse model. Overall well written paper with excellent data and well supported conclusions.
Minor points
Lines 34: normally 30ies and 40ies is written 30s and 40s
Table 1: while an n of 8 is an excellent number for clinical samples is there any gender effect as the UC group has 5/3 male to female where the control group is 2/6?
Methods: Was gender noted and tracked for the mice? Is there any gender effect?
Results: Well presented
Conclusions: Al the conclusions are warranted and placed in appropriate context. I would Ideally like to see some discussion of gender as it is generally accepted there are some differences in IBD.
Author Response
Reviewer 2
Great use of human tissue, culture and mouse model. Overall well written paper with excellent data and well supported conclusions.
Minor points
Lines 34: normally 30ies and 40ies is written 30s and 40s
Table 1: while an n of 8 is an excellent number for clinical samples is there any gender effect as the UC group has 5/3 male to female where the control group is 2/6?
Methods: Was gender noted and tracked for the mice? Is there any gender effect?
Results: Well presented
Conclusions: Al the conclusions are warranted and placed in appropriate context. I would Ideally like to see some discussion of gender as it is generally accepted there are some differences in IBD.
We thank the reviewer for the kind conclusion and the suggestions. We agree with the reviewer and also were aware that in colitis gender might play a role. Thus, only male mice were used in this study. We added a note to make this clear in the methods section and the results part. In patients, we did not see obvious expression differences that were suggesting to be based on patient’s gender and thus did not discuss the role of gender differences in IBD in the context of MD3. However, for such effects, it might be necessary to analyze larger cohorts. As well as MD3 appears to be involved in regulatory processes it might be not excluded that also hormonal or other sex-specific regulation might be affected or could affect MD3.
Reviewer 3 Report
”MarvelD3 in inflammatory bowel diseases has attenuating effects indirectly stabilizing the intestinal barrier” By Weiß et al.
This study aims to delineate the functional role of the tight junction-associated MARVEL protein D3 in the intestine of IBD-patients, in a human colonic cell line and in MD3-overexpressing mice. The authors suggest a protective role of the protein, however demonstrating increased expression in ulcerative colitis, possibly induced by IL-13 via the IL13Rα1/STAT pathway. The main strength of the study is the combination of human, mouse and cell-line experiments, and the use of relevant methods. The conclusion about using specific anti-IL-13R α2-therapy is interesting.
Comments:
- In the title, MarvelD3 is said to have an effect in inflammatory bowel diseases. However, nothing in the results indicate an effect in Crohn’s disease, as MD3 was only increased in ulcerative colitis. In fact, there is no evidence that MD3 has an attenuating effect in human UC; the evidence was only found in DSS-treated mice. I suggest that the title is changed accordingly.
- In general, it feels like the study is made up by three parts (IBD-patients, a human colon cell-line and mouse studies) that each are interesting separately, but are not really discussed collectively: for example, it should be clarified if IL-13 is increased in UC but not in CD, and if this could explain the differences between these two patient groups in MD3 expression. What is the role of IL-13 in DSS-induced inflammation? How come the expression of MD3 in human UC is increased when it is supposed to be protective? Etc.
- The patient characteristics are not very well described. The scoring systems should be explained so that the reader can understand if the patients had mild, moderate or severe disease. It would also be of interest to know about any treatments as this can affect cytokine levels, and in the end the tight junction expression.
- Did you see any difference in MD3-expression within the UC-group depending on disease severity that could indicate if the protein has an protective role?
Minor:
- The paper would benefit from proof reading as there are typos here and there, one example in the discussion (line 352): ”…which however was significant”, where I suppose that ”not” was missing.
- Figure 3B: What is Ctrl? Is that IL-13 without inhibitor? Please clarify.
Author Response
Reviewer 3
This study aims to delineate the functional role of the tight junction-associated MARVEL protein D3 in the intestine of IBD-patients, in a human colonic cell line and in MD3-overexpressing mice. The authors suggest a protective role of the protein, however demonstrating increased expression in ulcerative colitis, possibly induced by IL-13 via the IL13Rα1/STAT pathway. The main strength of the study is the combination of human, mouse and cell-line experiments, and the use of relevant methods. The conclusion about using specific anti-IL-13R α2-therapy is interesting.
Comments:
- In the title, MarvelD3 is said to have an effect in inflammatory bowel diseases. However, nothing in the results indicate an effect in Crohn’s disease, as MD3 was only increased in ulcerative colitis. In fact, there is no evidence that MD3 has an attenuating effect in human UC; the evidence was only found in DSS-treated mice. I suggest that the title is changed accordingly.
Considering the request, we changed the title to "MarvelD3 is upregulated in ulcerative colitis and has attenuating effects during colitis indirectly stabilizing the intestinal barrier".
- In general, it feels like the study is made up by three parts (IBD-patients, a human colon cell-line and mouse studies) that each are interesting separately, but are not really discussed collectively: for example, it should be clarified if IL-13 is increased in UC but not in CD, and if this could explain the differences between these two patient groups in MD3 expression. What is the role of IL-13 in DSS-induced inflammation? How come the expression of MD3 in human UC is increased when it is supposed to be protective? Etc.
The reviewer is right that we could not directly see protective effects in the patients. However, this underlines the importance of the animal model, where a single factor could be genetically influenced and allows to study its effect. Based on the finding of amelioration one could speculate about the ameliorating effect in the human, where we unfortunately cannot modify one TJ protein specifically. The upregulation might be interpreted as a counter-regulatory effect, which is no new mechanism. This is also discussed for claudin-2. Claudin-2 is upregulated in IBD and because it forms a water- and cation-selective paracellular channel, its upregulation leads to the symptom of leak-flux diarrhea. Although for a long time this was assumed to be a negative effect, mice that overexpress claudin-2 develop less severe colitis (Ahmad et al., 2014 à citation 52) and supported the hypothesis called "enteric tears". We added this to the discussion as it might help to understand that upregulation as we observe for MD3 may have indeed ameliorating, counter-regulatory effects.
As requested, in the discussion we clarified the relation of the three experimental approaches to each other.
- The patient characteristics are not very well described. The scoring systems should be explained so that the reader can understand if the patients had mild, moderate or severe disease. It would also be of interest to know about any treatments as this can affect cytokine levels, and in the end the tight junction expression.
The two scoring systems used for the UC and the CD patients are shortly introduced in the Methods section describing the patients. Relevant citations explaining the respective scoring systems in more detail are also mentioned in this section. We added for both explanations of the meaning of the different grades, which might help the reader to understand the ranges presented in the table and show that mild, moderate and severely affected patients were analyzed. However, we did not see that MD3 expression correlated with disease activity.
The reviewer is right, the effect of treatments would be interesting to study in order to see how they affect expression of cytokines and finally TJ protein expression. However, for this a long-term study should be performed to see time- and treatment-dependent changes, as also individual expression differences and therapy response need to be considered. Though being of high interest, this was beyond the scope of our present study.
- Did you see any difference in MD3-expression within the UC-group depending on disease severity that could indicate if the protein has an protective role?
No, we did not see a correlation of MD3 expression and disease severity. This might be due to low numbers of patients analyzed. One should take in account that MD3 is not the only factor playing a role in disease severity. Thus, much bigger cohort studies might be necessary to elucidate some effects when defining subgroups that for example might be matched in expression levels of other relevant proteins.
This also supports why (genetically modified) mouse studies where the background within the breeding population is very similar allow to find effects of one protein.
Minor:
- The paper would benefit from proof reading as there are typos here and there, one example in the discussion (line 352): ”…which however was significant”, where I suppose that ”not” was missing.
We carefully revised the manuscript and corrected several typos.
- Figure 3B: What is Ctrl? Is that IL-13 without inhibitor? Please clarify.
Yes, control is the procedure without any inhibitor pretreatment, which again led to increase of MD3 expression, when IL-13 was added. We added an explanation to the figure legend.